# Use of social media platforms by migrant and ethnic minority populations during the COVID-19 pandemic: a systematic review

Lucy Pollyanna Goldsmith ![ORCID] ,[1,2] May Rowland-Pomp,[1] Kristin Hanson,[3] Anna Deal,[1,4] Alison F Crawshaw ![ORCID] ,[1] Sally E Hayward ![ORCID] ,[1] Felicity Knights,[1,2] Jessica Carter ![ORCID] ,[1] Ayesha Ahmad,[5] M Razai ![ORCID] ,[2] Tushna Vandrevala,[6] Sally Hargreaves ![ORCID] [1]

LPG, MR-P and KH are joint first authors.

For numbered affiliations see end of article.

**Correspondence to**
Dr Sally Hargreaves;
s.hargreaves@sgul.ac.uk

## ABSTRACT

**Objective** Migrants and ethnic minority groups have been disproportionately impacted by COVID-19 and have lower levels of vaccine uptake in some contexts. We aimed to determine the extent and nature of social media use in migrant and ethnic minority communities for COVID-19 information, and implications for preventative health measures including vaccination intent and uptake.

**Design** A systematic review of published and grey literature following the Preferred Reporting Items for Systematic Reviews and Meta-Analyses guidelines. We searched databases including Embase, Web of Science, PubMed NIH, CINAHL, facilitated through the WHO Global Research on COVID-19 database from 31 December 2019 to 9 June 2021.

**Eligibility criteria for study selection** Research reporting the use of social media by migrants and/or ethnic minority groups in relation to COVID-19.

**Data extraction** We extracted data on key outcomes, study design, country, population under study and sample size.

**Results** 1849 unique records were screened, and 21 data sources were included, including populations in the UK, USA, China, Jordan, Qatar and Turkey. We found evidence of consistent use of a range of social media platforms for COVID-19 information in some migrant and ethnic minority populations (including WeChat, Facebook, WhatsApp, Instagram, Twitter, YouTube), which may stem from difficulty in accessing COVID-19 information in their native languages or from trusted sources. Some evidence suggested circulating misinformation and social media use may be associated with lower participation in preventative health measures, including vaccine intent and uptake, findings which are likely relevant to multiple population groups.

**Conclusions** Social media platforms are an important source of information about COVID-19 for some migrant and ethnic minority populations. Urgent actions and further research are now needed to better understand effective approaches to tackling circulating misinformation, and to seize on opportunities to better use social media platforms to support public health communication and improve vaccine uptake.

## STRENGTHS AND LIMITATIONS OF THIS STUDY

⇒ Comprehensive systematic review methods were used, following Preferred Reporting Items for Systematic Reviews and Meta-Analyses guidelines.
⇒ Both published and grey literature were searched and papers from all countries and all regions were included, allowing available evidence to be synthesised.
⇒ We acknowledge the limited geographical scope of included studies, with no data available from low-income countries.

**Registration** This study has been registered with PROSPERO (CRD42021259190).

## INTRODUCTION

The pandemic has been accompanied by an infodemic, defined as an excess of information during a disease outbreak—including false or misleading information in digital and physical environments[1]—that makes it difficult to distinguish reliable information from misinformation including disinformation (deliberate misinformation) and conspiracy theories. The WHO highlights that in all communities, infodemics cause 'confusion and risk-taking behaviours that can harm health…it leads to mistrust in health authorities and undermines the public health response, and can intensify or lengthen outbreaks'.[1] The rapid expansion of internet and social media use, in particular, in recent years (including platforms such as Twitter, WhatsApp and YouTube; table 1) has meant that both useful and potentially harmful health information can spread rapidly. Although social media can be used to disseminate factual, appropriate and useful information, a large proportion of the most popular

**Table 1** Popular social media platforms statistics from Statistica[71]

| Platform | Primary feature | Country of origin | Organisation | Users | Notes |
|---|---|---|---|---|---|
| YouTube | Online video sharing and social media platform. Free to use. | USA | Google | Approximately >2 billion monthly | Searchable. |
| WhatsApp | Messaging platform, allows users to send text messages and voice messages, make voice and video calls, and share images, documents, user locations, and other content. Free to use. | USA | Meta | Approximately >2 billion monthly | Not searchable. Groups can have 512 users in them. |
| Instagram | Photo and video sharing social networking site. Free to use. | USA | Meta | Approximately 1 billion monthly | Searchable. Some content is limited to connections only. |
| Facebook | Social networking service, allows messaging, image and video sharing, marketplace online shopping, live video sharing. Free to use. | USA | Meta | 2.89 billion active monthly | Searchable. Some content is limited to connections only. |
| WeChat | Instant messaging, social media, mobile payment. Free to use. | China | Tencent Holdings Limited | 1.25 billion monthly | Searchable. Some content is limited to connections only. |
| TikTok (Known in China as Douyin) | Video sharing focused on short form videos (15 s to 3 min). Free to use. | China | ByteDance | 837 million monthly active | Searchable. Some content is limited to connections only. |
| Snapchat | Photo sharing multimedia app with video features. Free to use. | USA | Snap | 347.3 million monthly active | Searchable. Some content is limited to connections only. |
| Twitter | Microblogging focused on short messages known as 'tweets'. Live chat event function Tweetchat. | USA | Twitter | 330 million monthly active | Searchable. Some content is limited to followers only. |

COVID-19 videos on YouTube, for example, have been found to contain misinformation, or no factual information, reaching millions of people worldwide.[2 3] YouTube is considered a major platform for information concerning the control of COVID-19, but most COVID-19 videos were of 'undesirable quality' containing few government/ public health recommendations according to a recent study.[4] A review of YouTube videos on general vaccination found 65% expressed antivaccination sentiment,[5] with antivaccine posts more likely to be recirculated on Twitter.[2] The spread of misinformation and disinformation has been highlighted as a major risk to ending the COVID-19 pandemic—including undermining trust in vaccines[6]—with researchers highlighting links between misinformation on social media and public doubts around vaccine safety, self-reported compliance with public health guidelines, and intent to vaccinate.[7 8]

Although social media platforms are commonly used in the general population, and patterns of use are complex across different population groups;[8 9] some migrant and ethnic minority groups—who may experience barriers to accessing health information and health systems— may be more reliant on social media and the internet

as a source of health information. These communities may also draw on diaspora media as a source of health information.[10] The COVID-19 pandemic has disproportionately impacted and exacerbated inequalities faced by migrants, and ethnically diverse communities—ethnic minority groups (including some migrant populations)— were at higher risk of contracting, being hospitalised with and dying from COVID-19.[10–15] They are also more likely to be vaccine hesitant—with lower take-up of preventative health measures, such as vaccines, noted in some groups due to a range of personal, societal and physical barriers.[12 14 16] Some migrant and ethnic minority communities may be more exposed to social media misinformation because of access barriers to accurate information (eg, from official government sources),[17 18] due to restricted eligibility and access to services, language barriers and low health literacy. However, little is known about the extent and nature of social media use in these populations, nor the impact that social media use has had on preventative health measures during the pandemic, including COVID-19 vaccine uptake. In addition, there is an opportunity now to explore the extent to which social media platforms could be better used to

support information sharing and promote public health messaging in marginalised communities during the pandemic and beyond.

We therefore did a systematic review to explore and assess the extent and nature of social media use by migrant and ethnic minority groups to access COVID-19 health information, the extent to which misinformation on social media may have influenced views about COVID-19 preventative measures including vaccination intention and uptake, and to explore good practice.

## METHODS

### Search strategy

The review[19] followed Preferred Reporting Items for Systematic Reviews and Meta-Analyses guidelines.[20] The study protocol is available from the International Prospective Register of Systematic Review (PROSPERO) database (CRD42021259190). A Boolean search strategy was developed containing terms relating to migrants, ethnic minorities, COVID-19, social media and misinformation (see online supplemental file 1). We included papers covering any prevention topic, including social distancing, hand washing, mask wearing, testing, isolation, test and trace activities, and vaccination. We searched the following databases: Embase, Web of Science, Oxford Academic Journals, PubMed NIH, Clinical Trials, China CDC MMWR, CDC reports, ProQuest Central (Proquest), CINAHL, Africa Wide Information (Ebsco), Scopus, PsycInfo, CAB Abstracts, Global Health, J Stage, Science Direct, *Wiley Online Journals*, *JAMA Network*, *British Medical Journal*, Mary Ann Liebert, *New England Journal of Medicine*, Sage Publications, Taylor and Francis Online, Springer Link, BioMed Central, MDPI, ASM, PLOS, *The Lancet*, Cell Press, and preprint sites chemRxiv, SSRNbioRxiv and medRxiv. This was facilitated through the WHO Global Research on COVID-19 database. We searched records from the date the WHO was first informed of COVID-19, 31 December 2019,[21] to 9 June 2021(https://search.bvsalud.org/global-literature-on-novel-coronavirus-2019-ncov/). The WHO COVID-19 Database[22] is a daily updated multilingual resource of all literature (peer-reviewed literature, preprints and grey literature) pertaining to COVID-19.

Records were imported to Rayyan QCRI.[23] Both title and abstract screening and full text screening were conducted independently by two reviewers (MR-P and LPG) using Rayyan QCRI.[23] Additional relevant papers and grey literature (eg, from third-sector organisation websites) were identified using hand searching including backwards and forwards citation tracking.

### Selection criteria and primary outcomes

Papers reporting the use of social media platforms and implications for preventative health measures and vaccination intent of migrants and/or ethnic minority groups to COVID-19 globally were eligible. All types of COVID-19 preventative health measures, including social distancing, hand washing, mask wearing, testing, isolating, tracing close contacts of people who have COVID-19, alongside preventative measures based on misinformation were included. To include all available evidence, all types of scientific articles, reports and commentaries, editorials, correspondence letters were eligible for inclusion. Social media platforms were defined as any medium whereby content (including images, videos and messages) is circulated to the general public and may include YouTube, Facebook, Twitter, TikTok and Snapchat. 'Migrants' were defined as foreign-born, residing outside of their country of birth. An ethnic minority group was defined as a group of people with a shared culture, tradition, language, history, living in a country where most people are from a different ethnic group, and will include migrants/foreign-born populations alongside individuals born in the host country. Where studies reported a general population sample, results about migrant/ethnic minority groups within that sample were eligible for inclusion. No papers were excluded based on language or geographical origin. Studies were excluded if it was not possible to determine whether individual(s) in the population studied were migrants or from an ethnic minority group.

### Data extraction, critical appraisal and synthesis

Data extraction was completed independently by two researchers (MR-P and LPG) using a piloted, structured data extraction sheet in Microsoft Excel. Fields in the data extraction sheet included author and year, dates for data collection, location of study, location of population of interest, whether qualitative methods were used, whether quantitative methods were used, study design, whether there was an intervention, the type of intervention, methodology, population of interest, further information about population, sample size, type of social media, misinformation type, participant recruitment strategy, and all outcomes. Outcomes were extracted as reported. Risk of bias was assessed independently by two researchers (LPG, MR-P) using the Quality assessment for Survey Studies in Psychology for Surveys[24] for quantitative studies. The 20 items on this scale can be rated as 'yes', or 'no', 'not stated clearly', or 'not applicable'. Scores are calculated by dividing the 'yes' answers by the total number of applicable items, with scores over 70% indicating 'acceptable' quality. The Critical Appraisal Skills programme (CASP) checklist was used for qualitative studies.[25] The 10 items can be rated 'yes', 'can't tell' or 'no'. We rated the CASP by dividing the 'yes' answers by the total number of applicable items, with a score of over 60% indicating 'acceptable' quality. We did not exclude any papers on the basis of quality. The selection of risk of bias rating instruments was finalised once we had a complete list of the type methods used in the included studies. We used a mixed-methods[26] narrative synthesis[27] approach, synthesising the qualitative and quantitative data together by theme.

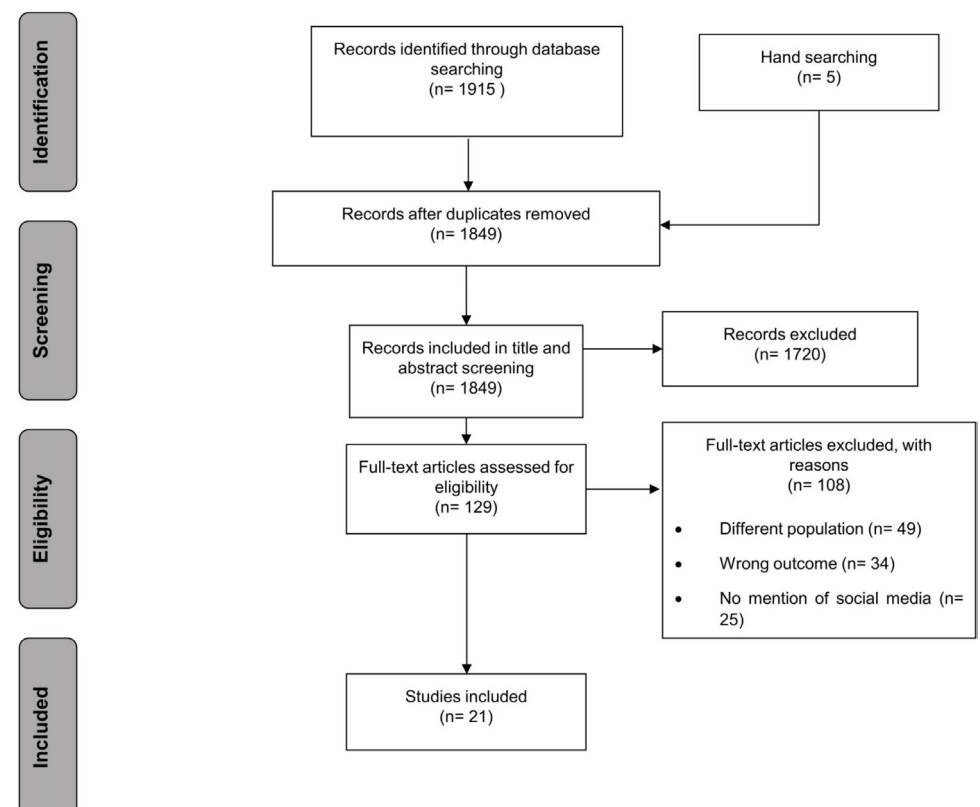

**Figure 1** Preferred Reporting Items for Systematic Reviews and Meta-Analyses (PRISMA) diagram.

### Patient and public involvement

Patients and/or the public were not involved in the design, conduct, reporting or dissemination plans of this research.However, three of the authors are from ethnic minority groups.

### RESULTS

#### Overview of data sources

Following de-duplication, 1849 unique data sources were identified and screened and ultimately 129 were full-text screened. Twenty-one data sources were included in the final analysis (figure 1). Six studies were conducted in the UK,[28–33] two were jointly conducted in the UK and USA.[34 35] An additional eight studies were conducted in the USA,[36–43] and one each in China,[44] Jordan,[45] Qatar[46] and Turkey.[47] Eight studies reported on migrants,[31 33 36 44–48] including migrants in the host countries of China,[44] Jordan,[45] Qatar,[46] Turkey,[47] and the USA[36] and UK,[31 33] and one study involved predominantly migrants from Venezuela residing in other countries.[48] Nine studies reported about a specific ethnic minority or group (Latino individuals,[36 38 40 41] black American citizens,[39 43] Jain community members[32] and Syrian migrants).[45 47] Seven studies reported about ethnic minority groups generally.[28–30 34 35 37 42] A survey design was the most common design, used in half of included studies.

Characteristics of included studies are presented in table 2, including the risk of bias assessment scores. Quality scores ranged from 76% to 90% for included papers, suggesting acceptable quality of all included data sources where quality assessment was applicable. The most common shortcomings for studies related to reporting about the ethics and participants. None of the included studies were preprints.

Online supplemental material 2 shows the geographical location of data sources, highlighting the absence of published and unpublished data on this topic from most regions of the world.

#### Use of social media platforms as a source of information about COVID-19

For some migrants and ethnic minority groups, consistent use of social media platforms for sharing and receiving COVID-19-related health information was reported in several included studies.[33 38–40 44–48] Figure 2 highlights quantitative data sets showing use of social media for information about COVID-19. Social media was reported to be the preferred source of information about COVID-19 for international migrants in China (WeChat was used by 94.5% of respondents for COVID-19 information).[44] Among 389 Syrian refugee mothers in Jordan,[45] Facebook and WhatsApp were the main sources of information for 87% and 69% of respondents, respectively, for COVID-19 information; with 21% indicating that they accessed information from professional databases or government websites, and 53% via television (this survey was circulated via Facebook and WhatsApp). Migrants from Venezuela (residing in numerous countries) reported Facebook and WhatsApp were their two primary sources of information

**Table 2** Characteristics of included studies

| Included study | Location of study | Study design | Population under study | Type of publication | Main topic of the paper | Sample size | Quality rating* |
|---|---|---|---|---|---|---|---|
| Alabdulla[46] | Qatar | Cross-sectional survey | Non-Qatari residents | Peer-reviewed journal | Vaccine hesitancy | 7821 | 76%† |
| Allington (a)[30] | UK | Cross-sectional survey | Non-white ethnic groups | Peer-reviewed journal | Vaccine attitudes, trust and COVID-19 information source as predictors of vaccine hesitancy | 4343 | 82%† |
| Allington (b)[35] | UK | Cross-sectional survey | Non-white ethnic groups | Peer-reviewed journal | Media usage predicts intention to be vaccinated against COVID-19 | 8988 | 89%† |
| Behbahani[36] | USA | Organisational case study | Latino migrants | Peer-reviewed journal | Helping vulnerable migrant populations in the COVID-19 crisis | N/K | NA† |
| Campos-Castillo[37] | USA | Cross-sectional survey | Non-white ethnic groups | Peer-reviewed journal | Racial and ethnic digital divides in posting COVID-19 content on social media | 10510 | 88%† |
| Cervantes[38] | USA | Qualitative interviews with thematic analysis | Low-income Latino individuals | Peer-reviewed journal | Experiences of Latinx individuals hospitalised for COVID-19—misinformation and disbelief | 60 | 90%‡ |
| Chandler[39] | USA | Qualitative interviews with thematic analysis | Black women (18–31 years) | Peer-reviewed journal | Evaluating the perspectives and sources of information of black women about COVID-19 | 15 | 90%‡ |
| Crawshaw[31] | UK | Evidence synthesis linked to outputs from participatory workshops with migrants | International migrants | Peer-reviewed journal | Vaccine hesitancy and barriers to COVID-19 vaccination in migrants | N/K | N/A§ |
| Despres[40] | USA | Organisational case study | Latino community living in America | Peer-reviewed journal | A digital content curation model to challenge the inequitable impacts of COVID-19 on US Latinos | N/K | N/A§ |
| Danish Refugee Council (DRC)[47] | Turkey | Cross-sectional survey | Syrian refugees in South-East Turkey | Research-based needs assessment report | The impact of COVID-19 on refugees in South-East Turkey | 774 | 82%† |
| Hamadneh[45] | Jordan | Cross-sectional survey | Syrian refugee mothers | Peer-reviewed journal | Knowledge and attitudes about COVID-19 among Syrian refugee women in Jordan | 389 | 78%† |
| Lockyer[28] | UK | Qualitative interview; reflective thematic analysis | People from different ethnic groups in Bradford | Peer-reviewed journal | COVID-19 misinformation and vaccine hesitancy in context | 20 | 90%‡ |

Continued

**Table 2** Continued

| Included study | Location of study | Study design | Population under study | Type of publication | Main topic of the paper | Sample size | Quality rating* |
|---|---|---|---|---|---|---|---|
| Loomba[34] | UK and USA | Randomised controlled experiment | Other ethnic groups than white | Peer-reviewed journal | The impact of COVID-19 vaccine misinformation on vaccination intent | 4000 (UK) 4001 (USA) | N/A§ |
| Moyce et al[41] | USA | Qual interviews narrative synthesis | Latino individuals | Peer-reviewed journal | Perceptions of COVID-19, news about COVID-19 and approaches to protecting health | 14 | 90%‡ |
| Paul[29] | UK | Repeated measures survey; cohort study | Other ethnic groups than white | Peer-reviewed journal | Attitudes towards COVID-19 vaccines, vaccine intent and implications for public health messaging | 32361 | 89%† |
| Regional interagency coordination platform (R4V)[48] | Any host county for migrants from Venezuela | Cross-sectional survey | Predominantly migrants from Venezuela | Research-based report | The difficulties encountered by refugees and migrants in the COVID-19 infodemic. Misinformation and vaccine hesitancy | 334 | 90%‡ |
| Vekemans[32] | UK | Organisational case study | Jain community members | Peer-reviewed journal | The relocation of the Jain community into the digital realm during the COVID-19 pandemic | 25000 estimate | N/A§ |
| Viswanath[42] | USA | Cross-sectional survey | Non-white ethnic groups | Peer-reviewed journal | Individual and social determinants of COVID-19 vaccine uptake | 1012 | 78%† |
| Wang[44] | China | Cross-sectional survey | International migrants | Peer-reviewed journal | COVID-19 knowledge, attitudes and sources of knowledge among international migrants in China | 1426 | 78%† |
| Woko[43] | USA | Cross-sectional survey | Black American citizens | Peer-reviewed journal | The role of beliefs and trust in COVID-19 information sources in low COVID-19 vaccination intention among black Americans | 1074 | 77%† |
| Deal[33] | UK | Qualitative in-depth interview study | Precarious migrants (asylum seekers, undocumented migrants, refugees) | Peer-reviewed journal | Action points to promote the equitable uptake of COVID-19 vaccinations for precarious migrants | 32 | 90%‡ |

*Scores were calculated on both scales by dividing the 'yes' answers by the total number of applicable items.
†Quality assessment for Survey Studies in Psychology for Surveys (Q-SSP) Checklist for surveys.
‡Critical Appraisal Skills Programme (CASP) checklist.
§N/A=not applicable due to research item design.

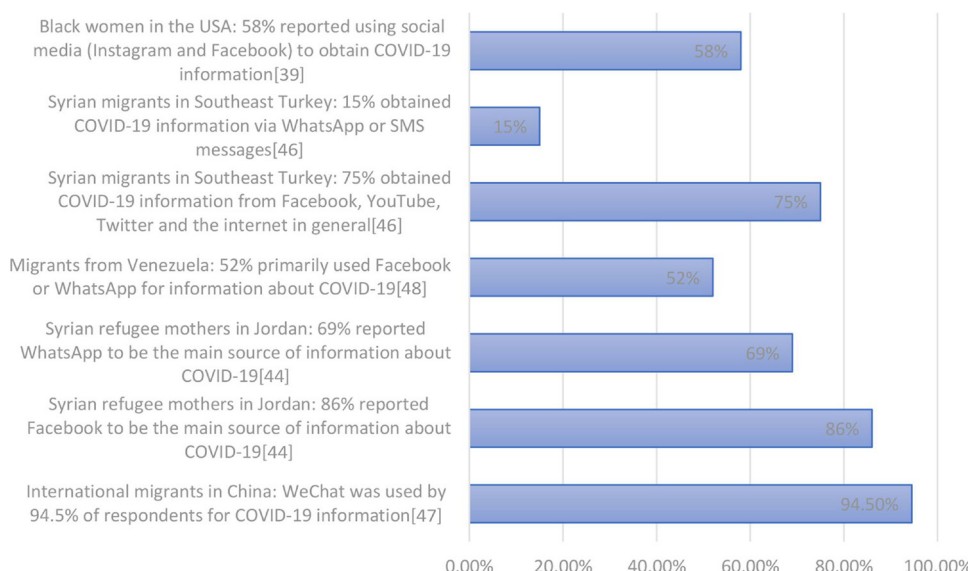

**Figure 2** Data on use of social media platforms as a source of information about COVID-19.

about COVID-19 in a survey of 334 migrants.[48] A survey of 774 refugee households in South-East Turkey[47] found the majority (75%) obtained COVID-19 information from Facebook, YouTube, Twitter and the internet in general, 15% via SMS/WhatsApp messages, followed by radio/TV (64%) and members of their community/family (34%); only 10% reported getting information from non-governmental organisations (NGOs)/United Nations sources. This study concluded that the heavy reliance on social media for information may expose a sizeable proportion of refugee households to fake or inaccurate information. In a US study of black women aged 18–31 years, 58% of respondents reported using social media (Instagram and Facebook) to obtain COVID-19 information.[39]

Participants from the US Latino community described relying on social media for information about the pandemic.[38] In Qatar, migrants reported they preferred to find out about COVID-19 using their own personal research or searching for information, including using social media as a source.[46] A study of precarious migrants (asylum seekers, undocumented migrants) in the UK found many relied on social media (WhatsApp groups, Facebook) for information on the pandemic and the ongoing vaccination programme.[33]

A key theme emerging in one UK study of ethnic minority groups[28] was that the 'avalanche' of information surrounding COVID-19 had led to interviewees feeling overwhelmed and confused: participants reported using a variety of sources of information, including TV, radio, news stations in Pakistan, India, Slovakia and Poland, online newspapers, Facebook, WhatsApp, Twitter, Google, and medical journals. A number of these participants said they dismissed some stories encountered on WhatsApp and Facebook; however, the sheer volume of messages coupled with the fact that people they trusted were sharing them, proved difficult to ignore, with participants raising concerns about how quickly social media

stories were shared. One study exploring the views of US Latinos reported that they consulted national and local news reports for information about COVID-19 and many reported that they got their news from Spanish-language news due to difficulty in understanding news in English; some received their news from social media sources, including Facebook, but expressed caution around messages from social media as there was no way to ensure the accuracy of the reports.[41] Language barriers were also reported among the Syrian refugee population in South-East Turkey who typically prefer information in Arabic,[47] as well as the Latino population in the USA,[36 38 40] for people from a range of ethnic groups in Bradford[28] and for international migrants in another study.[31]

According to one study, members of ethnic minority groups were also more likely to post COVID-19 content on social media than white individuals,[37] with respondents who identified as black (OR 1.29, 95% CI 1.02 to 1.64; p=0.03), Latino (OR 1.66, 95% CI 1.36 to 2.04; p<0.001) or other races/ethnicities (OR 1.33, 95% CI 1.02 to 1.72; p=0.03) having higher odds than respondents who identified as white of reporting posting COVID-19 content on social media.

### Drivers of social media reliance

Studies reported that some migrant and ethnic minority groups turned to social media as a result of a need for connection and to acquire accessible information from people they considered to be reliable sources.[32 41] For the Latino community in the USA, faith and community bonds were valued ways of coping with the difficulties of the pandemic which included feelings of social isolation, stress and uncertainty and—according to one study—social media facilitated these connections in a virtual space.[41] The Jain community in London used social media to communicate news and knowledge about COVID-19 and stay connected online, with events moving to a virtual

space; individuals reportedly benefited from and were grateful for this community use of social media.[32]

Several studies highlight concerns that some migrant and ethnic minority groups were unable to find official information in their host country in their native language about various aspects of COVID-19, hence their reliance on social media.[28 31 33 44 47] For example, a UK study of precarious migrants (asylum seekers, undocumented migrants) reported that those feeling most abandoned or scared due to a lack of understandable, clear official information in the early stages of the pandemic were more likely to rely on word of mouth or social media (WhatsApp groups, Facebook) for information, including around the vaccination programme.[33] One study of international migrants in China (94.5% of whom preferred social media for news about COVID-19) had lower rates of correct knowledge about COVID-19 compared with rates reported for Chinese residents.[45] The authors speculate that this might be due to a lack of available public health information in a range of languages.

Other studies showed positive associations with use of social media and access to information. One study highlighted that social media can support migrants to navigate the complex medicolegal context of their host countries by accessing information about public health measures and how to access medical help.[36] Social media use was associated with improved knowledge about COVID-19 and how to stay safe, in studies of Syrian refugee mothers[45] and US Latinos.[40] In another study specifically curated, culturally relevant digital content was considered to be an effective health promotion tool to share knowledge about practical actions to be taken to address the inequitable impact of the pandemic on US Latinos.[40]

## Misinformation and social media use

A summary of some of the key misinformation narratives identified in studies is provided in table 3. Some studies made links between social media and circulating misinformation in migrant and ethnic minority groups. For example, a UK cohort study found that belonging to an ethnic minority group and socioeconomic disadvantage was associated with exposure to misinformation about vaccines and mistrust in information about COVID-19.[29] A study of Syrian refugee mothers in Jordan, who reported receiving most of their COVID-19 information through social media, identified some erroneous beliefs about pregnancy, COVID-19 and breast milk.[45] A UK study among ethnic minority groups reported that participants encountered a range of misinformation, usually through social media sources and that vaccine hesitancy could be attributed to safety concerns, negative stories and personal knowledge, all of which had been amplified by recent exposure to misinformation via social media.[28] Myths identified included the idea that health professionals at the local hospital were injecting people with COVID-19 or killing people with the COVID-19 vaccine; there were wider beliefs reported about vaccines containing a microchip; making people infertile, or that

vaccines are being tested on ethnic minority individuals.[28] These participants described the dilemma of not knowing what to trust or who to listen to, including the videos/ posts that appeared to be from trusted professionals; therefore, they could not entirely dismiss negative stories circulating via social media and elsewhere.

In a study of 60 Latinx adults who had been hospitalised for COVID-19 in the USA, many of these adults reported that they relied on social media for COVID-19 recommendations.[38] They described a lack of information and the presence of circulating misinformation, with suspicion of the government and immigration departments reported as a common misinformation theme: 'some of us see (COVID-19) as a tactic for the government to access our documentation status and deport us'.[38] One Mexican man (age 45 years) in one US study[41] noted: 'When someone uploads something to Facebook then no-one believes in it 100%'; a Mexican woman (age 33 years) was also quoted as saying "(I get my information) well through the news, TV, Facebook and all of that…not everything I see is credible".

In a UK qualitative study,[28] participants who initially disregarded conspiratorial beliefs found it challenging to maintain their confidence that the rumours were untrue due to a number of factors: (1) Receiving many social media messages about them; (2) Receiving messages about them from trusted others; (3) Feeling anxious; and (4) Being under lockdown conditions at home. Participants expressed confusion about which story to trust, and ongoing difficulty identifying information as misinformation and dismissing it. Similarly, another study[39] reported that 79% of female black Americans interviewed stated that they were confused by the COVID-19 information they'd accessed from any source: 'Sometimes I feel unsure about the information that I'm receiving because it's a lot of different things about it. Everybody's not saying the same thing. So, I'm kind of unsure about what to believe'.[39]

## Social media impact on preventative health measures and vaccine intent

A small number of studies linked social media use with lower participation in preventative measures among migrants and ethnic minority groups. A UK/US survey study found vaccine hesitancy to be associated with informational reliance on social media and membership of an ethnic minority group.[30] A UK qualitative study reported that ethnic minority groups were influenced by antivaccine misinformation, including from social media.[28] A UK qualitative study of precarious migrants found that among 23 participants who were hesitant about receiving a vaccine some participants described fears around theories based on misinformation, often originating from social media or word of mouth, with many describing feeling conflicted about which information sources to trust.[33] Community leaders from African, Caribbean, Asian and Eastern Mediterranean migrant groups in London, UK reported substantial COVID-19

**Table 3** Examples of circulating misinformation on social media platforms relating to COVID-19, reproduced and compiled from Loomba et al[34] and Lockyer et al[28]

| Misinformation identified | Source | Engagement* | Reach† |
|---|---|---|---|
| "Scientists have expressed doubts over the effectiveness of a COVID-19 vaccine that has been rushed to human trials, after all the monkeys used in initial testing later contracted COVID-19." | Twitter | 1.59 K | 1.5 m |
| "The new vaccine for COVID-19 will be the first of its kind EVER. It will be an mRNA vaccine which will literally alter your DNA. It will wrap itself into your system. You will essentially become a genetically modified human being" | Twitter | 27 | 19.6 K |
| "They said it was just to flatten the curve. Now it's a battle for human survival." The only must-see action thriller for 2020. Starring: Bill Gates, Anthony Fauci, Chris Witty, Matt Hancock. Guest mask appearances: Clintons, Boris Johnson, Nicola Sturgeon, Joe Biden & Tedros. (Graphic featuring Mr. Bill Gates with the following quote.) "If we do a really good job with vaccines, we can reduce population by up to 15%. But if we create a worldwide pandemic first, killing people and making many of the survivors sterile, then create the vaccine, we may achieve the Georgia Guidestones first commandment!" | Twitter | 11 | 1.49 K |
| Something is very fishy about all this indeed. "A VIRUS WITH A 99.6% SURVIVAL RATE FOR PEOPLE UNDER 70 BUT THE ENTIRE WORLD NEEDS TO TAKE A VACCINE? I'M NO SHERLOCK HOLMES BUT SOMETHING'S FISHY ABOUT ALL THAT......." | Twitter | NK | 32.5 K |
| "Big Pharma whistle-blower: '97% of corona vaccine recipients will become infertile'" | Twitter | 6.95 K | 336 K |
| "I've been in Twitter jail for the last 12 hours for posting a link to a peer reviewed scientific study published in Vaccine showing that in military personnel prior receipt of the influenza shot increased COVID-19 risk by 36%. Censorship is vile & unAmerican." | Twitter | 25.1 K | 1.41 K |
| "So we know for a fact that the influenza vaccine worsens COVID-19 symptoms. So what are they mandating now? The influenza vaccine, of course." | Facebook | NK | NK |
| "PREPARING THE PROPAGANDA BLITZ. Yale University and the U.S. government are running clinical trials to develop propaganda messaging to persuade Americans to take experimental, genetically engineered, unlicensed, "Warp Speed," zero liability, expedited vaccines with limited short duration safety testing. Researchers compared reactions in 12 focus groups using "guilt, embarrassment, bravery, anger, trust" and "fear" to overcome vaccines hesitancy" | Instagram | 28.2 K | NK |
| ▶ COVID-19 is not real, it is an effort to control society<br>▶ COVID-19 has been manufactured by China or other governments for control purposes<br>▶ COVID-19 is caused by 5G<br>▶ COVID-19 has been invented to make people use contactless payments so that the government can track individuals<br>▶ COVID-19 testing gives so many false positives that it is ineffective and you should not self-isolate<br>▶ COVID-19 exists but is not as virulent as the government says it is<br>▶ If children test positive for COVID-19 during school hours, they can be taken away into care and will not be able to see their parents until they test negative<br>▶ The COVID-19 vaccine contains a chip that will track individuals, stop them travelling, and so on<br>▶ The COVID-19 vaccine will make people infertile and is an attempt to reduce the population, particularly targeted at people from BAME communities<br>▶ BAME people are being used as 'guinea pigs' to test out the COVID-19 vaccine<br>▶ The COVID-19 vaccine has been developed and approved too quickly and has not been fully tested<br>▶ The COVID-19 vaccine will negatively disrupt your natural immune system<br>▶ Herbal remedies will be more effective than the COVID-19 vaccine | Multiple platforms/ unknown | NK | NK |

*Engagement measures the number of likes and retweets.
†Reach measures the number of followers and thus potential audience size.
NK, not known.

vaccine hesitancy due to misinformation circulating on social media and word of mouth combined with a lack of accurate, translated and clear guidance.[31] Similarly, in a US qualitative study of Latino adults, some participants reported encountering a lack of knowledge accompanied by misinformation on social media causing them to dismiss preventative measures.[38] Another US study among Latino people reported that social media acted as a potential deterrent for following some public health measures to prevent infection by encouraging rule-breaking behaviour through socially normalising such behaviour by enabling people to observe the negative, guideline-breaking behaviours of others in social media posts.[41]

On the other hand, a large (8001 participants) US/UK randomised controlled experiment[34] found no significant differences in the response of different ethnic groups to misinformation in relation to vaccine intent. A large US/UK study[35] found membership of an ethnic minority group was associated with reduced vaccine intention, a relationship which was significant in three out of four studies (p<0.001, n=3890; p=0.017, n=1663; p<0.001, n=2237). The relationship persisted even when use of legacy (print and broadcast media) and frequency of use of social media was controlled for. High levels of social media use was not associated with vaccine intent in any of the three studies exploring this relationship; however, high information reliance on social media was significantly associated with negative vaccine intent (p=0.028, n=2237), suggesting a reliance on social media for information can make users vulnerable to misinformation. This study did not include interaction terms between ethnicity and information reliance on social media, which could have indicated whether the effect of information reliance on social media on vaccine intent differs by ethnicity.

### Good practice in promoting information and countering misinformation

Evidence suggests the important role of strong connections with the local community to identify and counter misinformation and rumours by trusted and valued sources of information. Most studies recommended improving the accessibility of public health information for migrant and ethnic minority communities.[29 31 36 38 39 41–44 48] For example, providing public health information in the media channel preferred by that group,[38] in multiple languages,[28] and using local, trusted voices delivering specific and targeted messages to counter fake news.[28 38] A strong interest in online, personalised information was identified.[40 41] Where social media was used to share personalised and culturally tailored public health information, it has a positive influence with good health knowledge, health-seeking behaviours and vaccine intent.[28–31 36 38–40 42–45 48] Studies indicated the need for culturally tailored health messaging to ensure equitable health knowledge for improving vaccine intent and health-seeking behaviours.[31 38–40]

More personalised means of health information communication was highlighted as a demand for informational reliance. A national US organisation which provides online health information tailored to the US Latino community found a high level of interest in their COVID-19 curated content, suggesting a strong demand for tailored and culturally relevant material.[40] In a new approach, 'virtual patient navigators', helpers working online, typically using messages to provide individually tailored health information, were made available to Latino migrants through a New York-based communication platform.[36]

Working through trusted sources was also emphasised. Providing accurate and tailored information about COVID-19 via trusted community members and organisations was suggested in a study of black women aged 19–31 years in the USA.[39] The study recommended that health professionals take an active role collaborating with the community to address inequities that black women are experiencing in the pandemic.[39] Participants in a randomised controlled study to explore the impact of misinformation on vaccine intent on different population groups reported finding videos on social media very engaging, especially when delivered in multiple languages by someone in a trusted profession (eg, doctor/teacher/nurse).[34]

Successfully countering myths was reported in a UK study where the local council rapidly responded to fake news circulating in the local population (eg, a rumour about children who test positive in school for COVID-19 being removed from the school and/or their parents until they test clear).[28] Videos to refute the myth were swiftly posted online in both Urdu and Punjabi, and these were reported to be effective by members of the local population.[28] Additional studies report successfully countering misinformation using a network of patient navigators[36] and community household surveys.[28] Social media use to communicate with family was also reported to be effective in challenging COVID-19 denial misinformation rumours through reporting of lived experience of COVID-19.[38]

## DISCUSSION

Among migrant and ethnic minority populations in the UK, USA, China, Jordan, Qatar and Turkey we found evidence of consistent use of social media for COVID-19 information, including via WeChat, Facebook, WhatsApp, Instagram, Twitter, YouTube, which may stem from a difficulty in accessing COVID-19 information in their native languages or from sources they trusted. There was some evidence of circulating misinformation and social media use associated with lower participation in preventative health measures, especially vaccination intent, and finding that will be undoubtedly generalisable to multiple population groups. This is a rapidly evolving field of research, and data are limited, but our work highlights the considerable importance of social media platforms as a source of information and misinformation

about COVID-19 for some migrant and ethnic minority populations during the pandemic. While we know social media is used by many people, and misinformation has been circulating widely in the general population, it may be the case that those excluded from national public health responses and/or who faced specific barriers to accurate public health information and support may have been disproportionately impacted. Urgent actions and further research are now needed to better understand use of social media platforms for health information in different population groups, find effective approaches to tackling misinformation, and to seize on opportunities to make better use of social media platforms to support public health communication and improve vaccine uptake globally. Furthermore, the findings highlight the crucial role of locally trusted sources in identifying and tackling misinformation, and underscores the benefits of disseminating personalised and culturally relevant health messages, including via social media.

This review is the first attempt to synthesise global studies exploring the use and impact of social media on migrant and ethnic minority populations during the COVID-19 pandemic. However, it is limited by the availability and quality of the data sets available. We acknowledge the limited geographical scope of included studies, with 16 of 21 studies focused on migrant and ethnic minority populations residing in the UK and USA and no data at all from low-income countries. It may be that the lower availability of research funding in low-income countries may explain the lack of studies from these countries. We acknowledge that definitions and terms pertaining to migrants and ethnic minorities and social media are used inconsistently in research; this is an ongoing challenge within the field, which has previously been evidenced in similar reviews, and may mean we have missed papers. This was mitigated against by searching the published and grey literature more widely. We also acknowledge that many of the surveys didn't formally report whether the social media feeds their responders were following were from 'official' sources (such as government or NGOs or from 'unofficial' sources), or from friends, relatives, or from accounts simply with many followers. However, the included data sets, and further qualitative work our group is currently doing in the UK, suggest they will be predominantly unofficial sources, with government public health teams in several countries very slow to make effective use of social media as a platform of communication at the start of the pandemic. A further limitation is that there were insufficient studies to reliably compare use of social media across type of migrants (refugees/asylum seeker, undocumented migrants), and future research should explore this. We acknowledge that migrants and ethnic minorities are a highly diverse group with a range of health and socioeconomic situations making it hard to generalise; however, there is evidence in several contexts that these populations may have been disproportionately impacted by the COVID-19 pandemic.[13 16 49]

The findings of our review have been confirmed by more recent studies. For example, a survey of migrants in Greece found their main source of information about the vaccine was via social media platforms and the internet in general, and that vaccine hesitancy was linked to a lack of adequate information and driven by fear, anxiety, exposure to negative news and misinformation.[50 51] In Turkey, a 2021 survey and feedback mechanism in refugee communities found information gaps, misconceptions and rumours about COVID-19 vaccines circulating mainly by word of mouth and on social media, undermining health information.[52] In a recent study of Venezuelan migrants in Latin America, 70% said they had access to a mobile phone, with the main communication channels being WhatsApp and Facebook, yet half said they felt uninformed.[53] We also found that some migrants and ethnic minorities used diaspora media as a source of COVID-19-related information during the pandemic, which merits further consideration in terms of understanding how to better engage these groups in preventative healthcare and vaccination, and has been previously reported in studies as influencing views and beliefs around vaccination.[54] Misinformation on social media correlated negatively with vaccine intention and our findings align with other research in this area and will undoubtedly be relevant to many other population groups.[2 7 33] A recent study among migrants and nationals in Qatar acknowledged 'personal research' via social media as important to them for seeking information about COVID-19 vaccines, underlining the key role social media has in influencing people's attitudes towards vaccine uptake.[46]

The European Centre for Disease Prevention and Control and other public health bodies have raised concerns around barriers to public health information among migrant populations and ethnic minority groups residing in Europe and other high-income countries during the pandemic.[13 14] Public health guidance in some countries was not initially tailored to the needs of migrant and ethnic minority groups.[18 55–57] A review of the availability of government-produced risk communications across Council of Europe member states in June 2021 found only 48% (23/47) of countries translated COVID-19 information into at least one migrant language, with information on testing or healthcare entitlements in common migrant languages only found in 6% (3/47), suggesting individuals not able to access information in the host country language may have been excluded to some extent from governments' public health messaging.[18] In Denmark, a series of qualitative interviews with migrants found that they felt uncertain regarding government guidance for COVID-19; although written material was translated into 19 languages, it was not effectively disseminated.[58] In Montreal, Canada, there were delays to publishing official multilingual fact sheets on COVID-19 guidelines, and information phone lines only operate in French and English; those who had arrived most recently, had lower language (French/English) ability or lower literacy, had more difficulty accessing local COVID-19 information.[59] Lack of English or French language at the time of immigration to Canada were associated with lower

rates of testing and higher per cent positivity for COVID-19 in recently arrived adult immigrants and refugees.[60] A study among refugees and migrants in deprived areas in Greece found that migrants may have difficulties understanding public health messaging due to cultural and language barriers.[18 56 61] Merely translating public health information is not likely to be sufficient; information needs to be tailored and targeted so it is conveyed in ways that resonate with the target population. A range of key resources and guidelines on risk communication and engagement strategies for COVID-19 public health responses, including vaccination, among marginalised populations globally are available, as well as a social media toolkit for healthcare practitioners (https://www.who.int/publications/m/item/a-social-media-toolkit-for-healthcare-practitioners—desktop).[62–64] However, it will be vitally important that the lessons learnt around communication of public health information to marginalised groups during the pandemic are meaningfully carried forward.

Where social media is used to share personalised and culturally tailored public health information, it has a positive correlation with good health knowledge, health seeking behaviours and vaccine intent.[40 45] Our research shows the need for culturally tailored health messaging to ensure equitable health knowledge and to improve vaccine uptake, by accurate public health messaging through trusted sources of information.[31 38–40] We make a number of recommendations for policy and practice, which include the need for systematic monitoring of information and attitudes circulating on social media,[65] as well as timely rebuttal of misinformation from trusted professionals (see box 1). Several resources are now available to support addressing misinformation about COVID-19 vaccines as well as fostering demand for vaccines.[66–68]

There is a stark lack of data on social media use from low-income and middle-income countries, which merits greater consideration as COVID-19 vaccination gathers pace in these contexts. Studies from high-income countries are also limited, with the majority of studies focused on the USA and UK. In addition, more evidence is needed to examine the role social media platforms play in positively or negatively influencing health behaviours such as vaccine intent and uptake for COVID-19 in all populations (including other excluded groups, for example, homeless, internally displaced people). Social media is an important source of health information for some migrant and ethnic minority communities and tackling misinformation needs to be done using this medium given the lack of trust in government messaging in some of these communities.[69] Our findings are consistent with those of others working in this field, which show that social media can have a crucial role in disseminating health information, tackling infodemics and misinformation.[4] There is an opportunity now to more effectively use social media to make vaccine intent desirable, appealing and normative among migrants and ethnic minority groups. There is an urgent need to address infodemic-related challenges in a rapidly changing information environment,

---

**Box 1    Key messages and recommendations**

⇒ Social media is an important source of health information for some migrant and ethnic minority communities, who may face barriers to accurate public health information, health and vaccinations systems. More evidence is urgently needed to examine the role social media platforms play in positively or negatively influencing health behaviours such as vaccine intent and uptake for COVID-19 in marginalised populations.

⇒ There is a stark lack of data on social media use from low-income and middle-income countries which merits greater consideration as COVID-19 vaccination gathers pace in these contexts.

⇒ More emphasis must be placed on exploring opportunities for sharing and transmitting accurate information via social media platforms, for example, to make vaccine intent desirable, appealing and normative.

⇒ Use of diaspora media by migrant populations, as a source of COVID-19-related information during the pandemic and for other health information, merits further research and greater consideration when designing and delivering public health interventions.

⇒ Proactively monitor social media platforms and other media sources to identify antivaccine sentiment, misinformation, fake news and rumours, and address them in real time.

⇒ There is a need to promote targeted and tailored health information to marginalised populations who face access barriers to health and vaccination systems, through preferred and trusted sources and channels of information including social media platforms, and to ensure investment in workforce and infrastructure to support this.

⇒ Engage with and involve communities in developing culturally specific messages and approaches, and support community-driven initiatives to identify at-risk groups, map local influencers, and define content for locally meaningful communication campaigns. Facilitate partnership working at the local level through involvement of diverse stakeholders and ensure community partners are recognised and reimbursed for their contributions and expertise.

⇒ Social media platforms should exercise more accountability and sign pledges to systematically track and remove harmful content that undermine public health measures, particularly during a public health crisis. The public must be empowered to identify and flag misinformation on social media.

⇒ Public health bodies and healthcare professionals should avoid a narrow focus on misinformation and a one-way communication of 'more accurate' information. They should seek to understand the underlying causes of exposure to and belief in misinformation including genuine knowledge void, access barriers and health literacy.

⇒ Lessons must be learnt around shortfalls in the communication of public health information to marginalised groups during this pandemic. Importantly, countries should gather and evaluate innovations and models of best practice in this area, which must be meaningfully carried forward to strengthen uptake of routine vaccinations and other public health interventions.

---

including real-time monitoring of social media messages and misinformation and the development of online tools to fight disinformation, with a focus on collecting stratified population data to enable targeted and tailored responses. Robust interventions relying on behavioural science to tackle misinformation using social media and evaluations are a plausible next step to address immunisation challenges for COVID-19 vaccines and also routine vaccines. Building trust in public health messaging,

identifying information gaps, finding innovative ways of disseminating health information, and detecting and responding to misinformation as it emerges remain a priority for public health.[69 70]

**Author affiliations**
[1]The Migrant Health Research Group, Institute for Infection and Immunity, St George's University of London, London, UK
[2]Population Health Research Institute, St George's University of London, London, UK
[3]Faculty of Health, Social Care and Education, Kingston University, Kingston-Upon-Thames, UK
[4]Faculty of Public Health and Policy, London School of Hygiene and Tropical Medicine, London, UK
[5]Institute of Medical and Biomedical Education, St George's University of London, London, UK
[6]Faculty of Health, Social Care and Education, Centre for Applied Health and Social Care Research, Kingston University, Kingston, UK

**Contributors** The study was conceptualised by SH, and the protocol and research question were developed by SH, LPG and MR-P. Searches were developed by MR-P and LPG, with input from SH and SEH. Screening was done by LPG and MR-P. Data extraction and analysis were done by LPG and MR-P, with input from SH and KH. The first draft of the manuscript was produced by LPG, MR-P and SH, and developed with KH and TV, who all contributed to interpretation of the results. All authors, including AD, AFC, SEH, FK, JC, AA and MR commented upon and approved the final manuscript. SH is guarantor of this study.

**Funding** This study was funded by the National Institute for Health Research (NIHR300072), the Academy of Medical Sciences (SBF005\1111), and the Medical Research Council (MRC/N013638/1). LPG, SH and AFC are funded by the NIHR (NIHR300072); AFC and SH are funded by the Academy of Medical Sciences (SBF005\1111). SH acknowledges funding from the Novo Nordisk Foundation (Mobility–Global Medicine and Health Research) and the WHO. AD and SEH are funded by the Medical Research Council (MRC/N013638/1). MR is funded by an NIHR In-Practice Clinical Fellowship (NIHR 302007). JC is funded by an NIHR In-Practice Clinical Fellowship (NIHR300290).

**Disclaimer** The views expressed are those of the author(s) and not necessarily those of the NHS, the NIHR, or the Department of Health and Social Care. The funder of the study had no role in study design, data collection, data analysis, data interpretation, or writing of the report.

**Competing interests** None declared.

**Patient and public involvement** Patients and/or the public were not involved in the design, or conduct, or reporting, or dissemination plans of this research.

**Patient consent for publication** Not applicable.

**Ethics approval** Not applicable.

**Provenance and peer review** Not commissioned; externally peer reviewed.

**Data availability statement** Data are available upon reasonable request. This is a systematic review of published data.

**ORCID iDs**
Lucy Pollyanna Goldsmith http://orcid.org/0000-0002-6934-1925
Alison F Crawshaw http://orcid.org/0000-0003-0450-7258
Sally E Hayward http://orcid.org/0000-0002-4105-0990
Jessica Carter http://orcid.org/0000-0001-9590-3146
M Razai http://orcid.org/0000-0002-6671-5557
Sally Hargreaves http://orcid.org/0000-0003-2974-4348

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
