## [Reviewer comments · BMJ Open]

ARTICLE DETAILS

TITLE (PROVISIONAL)	The use of social media platforms by migrant and ethnic minority populations during the COVID-19 pandemic: a systematic review
AUTHORS	Goldsmith, Lucy; Rowland-Pomp, May; Hanson, Kristin; Deal, Anna; Crawshaw, Alison; Hayward, Sally; Knights, Felicity; Carter, Jessica; Ahmad, Ayesha; Razai, M; Vandrevala, Tushna; Hargreaves, Sally

VERSION 1 – REVIEW

REVIEWER	Kaushal, Aradhna University College London, Behavioural Science and Health
REVIEW RETURNED	16-May-2022

GENERAL COMMENTS	Thank you for the opportunity to read and review this very important and interesting paper. I believe publication of this review will provide a valuable contribution to the field. The paper is very well written, however I do feel it suffers from too wide a scope, which might be addressed by re-minor structuring. It was sometimes difficult to follow and keep track of the results as we the paper deals with both ethnic minority groups and migrant groups. In addition, there is a large range of possible outcomes as this has been specified as just “public health messaging” but could include everything from social distancing and mask wearing, to testing and vaccinations – all of which have their own specific complexities in relation to social media. I have made some more specific comments and suggestions below which I hope you find helpful. Abstract 1) You mention that there are some positive effects but do not outline what these are here. I think this was just one study so perhaps omit from the abstract or briefly explain what this was. Introduction 2) This goes for the abstract too – I think the rationale could be more clearly outlined here. In the UK at least, ethnic minority groups (and perhaps migrant populations) were are higher risk of contracting covid-19 and were more likely to be hospitalised and die from it. They are also more likely to be vaccine hesitant. These facts together form a strong reason for understanding how/what health information reaches and is processed by these populations. 3) Page 4 line 10: Again, it would be helpful to highlight the usefulness of social media since you mentioned it. 4) Table 1. I like this table – it’s great for understanding and setting the context. The problem then is that more information is needed to fully understand the differences and similarities in the platforms e.g. Whatsap operates through phone numbers and the people in the groups/chats are generally friends and family who are known to each other in real life. Whereas Instagram information is based on
---

who you choose to follow. Some of the platforms are searchable and other are not so there is a difference between being presented information by a family member on whatsapp and doing your own research on Twitter. There may be some copyright issues with re-producing the table unless you have obtained permission.

5) Page 6 line 3. Some more detailed information would be helpful here and/or in the methods about what public health messaging you included and what you excluded i.e., social distancing, hand washing, mask wearing, testing, isolating, taking part in test and trace activities, vaccinations etc.

Methods

6) Looking at the search strategy, it doesn't look like you used subject headings or Mesh terms. Could you clarify if this was the case?

7) You say you searched from "inception to 9/6/21" Do you mean the inception of Covid-19 or all records available? Please provide the beginning of the date range for your search.

8) Can you justify why you chose to search pre-prints, and include editorials and letters? I think it's fine to do so but any included papers from pre-prints should be highlighted in your results as these are not peer-reviewed.

9) Please provide more details about how you searched the grey literature. Did you run the Boolean search via google or go to specific websites?

10) Please provide more details and a reference about the methodology used synthesise the data beyond "narrative synthesis". How did you go about synthesising the qualitative and quantitative data (separately or together)? – see Pluye and Hong (2014) for some more guidance on this.

11) Can you justify why you used the Q-SSP? Most of your included studies were surveys – was this chosen a priori or selected once you had an idea of the number of survey studies included?

12) Please explain which fields were in you data extraction form.

13) I know it's not possible to go back and include people but the fact that no members of the public were involved in the research is a limitation which should be acknowledged in the discussion section. Having this perspective would have been a strength of this study - you might want to consider a reflexivity statement and whether anyone in the research team is from an ethnic minority or migrant group if they feel it had any bearing on the choice of study design and interpretation of results.

14) Were any of the papers not in English? What was the process of translation and extraction for these papers?

Results

15) Page 8 line 44: Could you explain what the main shortcomings of the papers were e.g. what were the items which most commonly were not met on the checklist?

16) Table 2: More information can be included here:

a. as mentioned before the source of information – journal article, report, pre-print etc

b. the main topic of the paper e.g. vaccination, attitudes etc.

c. whether or not there was a comparison group and what it was

17) Table 2: Could you clarify if the surveys were cross-sectional?

18) Page 10 (Use of social media platforms as a source of information about Covid-19)–the information in this paragraph would be very nicely summarised as a graph.

19) Page 10 (Use of social media platforms as a source of information about Covid-19) – did you find any information about

	whether the sources of information were official vs unofficial? In the UK, Public Health England and the NHS undertook social media marketing on Twitter and even on Youtube by collaborating influencers. 20) Page 11 line 29: typo on member 21) Page 11 line 43: typo in first sentence 22) Table 3: Again -this may need permission for reproducing. I'd suggest summarising in one paragraph rather than having a table here. 23) Page 15 line 51-55: I read this a few times but struggled with understanding the meaning. Consider re-phrasing/clarifying. 24) Figure 1: the full-text articles excluded add up to 118 instead of 108. Is this because some articles were excluded for more than one reason? 25) Figure 2 is not necessary as it repeats the information presented in table 2. I can see you are highlighting the lack of research in low-income countries and more generally worldwide but this is sufficiently done in the text. Discussion 26) Typo on "tacking" 27) Please add a strengths and limitation section 28) Table 4 is not necessary. The key messages should be in the opening paragraph of your discussion. You can add a section to your discussion on the implications for policy and practice. Alternatively, the recommendations could work nicely as a figure/infographic.
--	---

REVIEWER	Requena-Mendez, Ana
	Instituto de Salud Global Barcelona
REVIEW RETURNED	02-Aug-2022

GENERAL COMMENTS	This article reviews the evidence about the use of social media platforms by migrant and ethnic minority communities for information about COVID-19. Also, it has tried to address the misinformation via these platforms and implications for health behaviours including vaccine intent and uptake. I think it is an interesting topic although data and studies are very scarce in this field. I also think the methodology is appropriate for a systematic review. I have several general comments:  1. Authors provide information on the use of social media platforms as a source of information about COVID-19. Are there specifications about the language for which these COVID-19 information is searched (country of birth language, host-country language, English....)? This would be very useful for future policy planning for example. 2. I would try to specify more in the section "use of social media platforms", which of these media are official sources of information (e.g., twitter or facebook from public health agencies, NGOs...) and which ones are coming from "unofficial sources". If studies do not differ this aspect, I would introduce it as a major limitation of such studies. 3. Are there any differences considering the type of migrants (refugees/asylum seeker, undocumented migrants...)? Authors recognize the limitation concerning the availability and quality of the datasets and also that that migrants and ethnic minorities are a highly diverse group with a range of health and socioeconomic
---

	situations making it hard to generalize. However, authors should compare different migrant groups, in particularly regarding the access to official information in their host country but also on the “social media impact on preventative health measures and vaccine intent”. Add it otherwise as a limitation. 4. Authors highlight the stark lack of data from low and middle-income countries. What are possible explanations for that? Publication bias? Are there other databases /search strategy (including non-scientific publications) that could add the perspective or context of low-middle income countries?. Minor comments: Page 13, Line 43. “Studies reported that some migrant ...” which studies? Please specify and also specify the targeted population Page 21, line 20. Why are not these recent studies not included in the review? I suggest updating the search strategy and to include reference 47-50 and 52. Page 23, line 29. “There is a stark lack of data on social media use from low and middle-income countries...”. I agree, but also from high income countries. Actually, the majority of studies were focused on USA and UK which also limits the generalization of the results also for migrant in high income countries. I would highlight it as a limitation of the study.
--	--

VERSION 1 – AUTHOR RESPONSE

Reviewer: 1
Dr. Aradhna Kaushal, University College London

Comments to the Author:

Thank you for the opportunity to read and review this very important and interesting paper. I believe publication of this review will provide a valuable contribution to the field. The paper is very well written, however I do feel it suffer from too wide a scope, which might be addressed by re-minor structuring. It was sometimes difficult to follow and keep track of the results as we the paper deals with both ethnic minority groups and migrant groups. In addition, there is a large range of possible outcomes as this has been specified as just “public health messaging” but could include everything from social distancing and mask wearing, to testing and vaccinations – all of which have their own specific complexities in relation to social media. I have made some more specific comments and suggestions below which I hope you find helpful.

Abstract

1) You mention that there are some positive effects but do not outline what these are here. I think this was just one study so perhaps omit from the abstract or briefly explain what this was.

Many thanks for raising this. We are very limited for word count in the abstract, so generally highlight the key message that there were positive and negative impacts, this is all expanded on in the results but no space in the abstract.

Introduction

2) This goes for the abstract too – I think the rationale could be more clearly outlined here. In the UK at least, ethnic minority groups (and perhaps migrant populations) were are higher risk of contracting covid-19 and were more likely to be hospitalised and die from it. They are also more likely to be vaccine hesitant. These facts together form a strong reason for understanding how/what health

information reaches and is processed by these populations.

This is a great suggestion, we have included an opening background section.

3) Page 4 line 10: Again, it would be helpful to highlight the usefulness of social media since you mentioned it.

Thanks, we've done this.

4) Table 1. I like this table – it's great for understanding and setting the context. The problem then is that more information is needed to fully understand the differences and similarities in the platforms e.g. WhatsApp operates through phone numbers and the people in the groups/chats are generally friends and family who are known to each other in real life. Whereas Instagram information is based on who you choose to follow. Some of the platforms are searchable and other are not so there is a difference between being presented information by a family member on WhatsApp and doing your own research on Twitter. There may be some copyright issues with re-producing the table unless you have obtained permission.

Thanks. We have added another column to Table 1 to clarify these issues. Interestingly, on WhatsApp there are a number of very large (up to 512 members) shared interest groups. We have clarified the preventative measures of interest in the methods section – this was all preventative measures, even ones based on misinformation.

We have generated this table ourselves from scratch, and quote the source of the stats so there are no copyright issues.

5) Page 6 line 3. Some more detailed information would be helpful here and/or in the methods about what public health messaging you included and what you excluded i.e., social distancing, hand washing, mask wearing, testing, isolating, taking part in test and trace activities, vaccinations etc. We included papers covering any prevention topic, including social distancing, hand washing, mask wearing, testing, isolation, test and trace activities and vaccination.

This is a good point, we covered all these topics and have added in the following text to the methods: "We included papers covering any prevention topic, including social distancing, hand washing, mask wearing, testing, isolation, test and trace activities and vaccination."

Methods

6) Looking at the search strategy, it doesn't look like you used subject headings or Mesh terms. Could you clarify if this was the case?

Many of the terms used in the search were also MeSH terms, but we did not limit ourselves to MeSH terms as these were all linked by the Boolean search term 'OR'. We searched using 'Title, abstract or subject' to leave the search as wide as possible. Our search strategy is available in the supplementary information.

7) You say you searched from "inception to 9/6/21" Do you mean the inception of Covid-19 or all records available? Please provide the beginning of the date range for your search.

We mean the inception of COVID-19; we've searched from 31/12/2019 as this is when COVID-19 was first reported to the WHO and we have updated our text accordingly. Many thanks for spotting this.

8) Can you justify why you chose to search pre-prints, and include editorials and letters? I think it's fine to do so but any included papers from pre-prints should be highlighted in your results as these are not peer-reviewed.

This was to include all available evidence. None of the included studies were ultimately pre-prints, however it was important to search these pre-print sites during the pandemic as evidence was becoming available very rapidly via these sites. We now make these points clearly in text so reader can tell where the evidence has come from.

9) Please provide more details about how you searched the grey literature. Did you run the Boolean search via google or go to specific websites?

The grey literature was included in the WHO COVID-19 database, an excellent resource pulling together both published and grey literature globally on COVID-19; we've now made this clearer in the text and supplementary materials.

10) Please provide more details and a reference about the methodology used synthesise the data beyond "narrative synthesis". How did you go about synthesising the qualitative and quantitative data (separately or together)? – see Pluye and Hong (2014) for some more guidance on this.

Thank you. We have clarified this in the text of the methods/page 8, adding in references.

11) Can you justify why you used the Q-SSP? Most of your included studies were surveys – was this chosen a priori or selected once you had an idea of the number of survey studies included?

This was selected once the number of studies which were surveys was established – this is now clear in the text.

12) Please explain which fields were in you data extraction form.

We've now included all the fields in the text of the paper in the methods section.

13) I know it's not possible to go back and include people but the fact that no members of the public were involved in the research is a limitation which should be acknowledged in the discussion section. Having this perspective would have been a strength of this study - you might want to consider a reflexivity statement and whether anyone in the research team is from an ethnic minority or migrant group if they feel it had any bearing on the choice of study design and interpretation of results.

This is a great point. We have clarified in the text that, three of the authors are from racially minoritized groups and three authors are migrants living in the UK.

14) Were any of the papers not in English? What was the process of translation and extraction for these papers?

We state in the paper "No papers were excluded based on language or geographical origin". However, the only eligible papers returned were all in English so there was no translation necessary. The search terms were in English, so any eligible foreign language papers may have been likely to have an abstract in English, but there were none.

Results

15) Page 8 line 44: Could you explain what the main shortcomings of the papers were e.g. what were the items which most commonly were not met on the checklist?

Good point, we have added this in. We have put this information on what was page 8, line 48, as this is where the quality scores are mentioned in the results.

16) Table 2: More information can be included here:

- a. as mentioned before the source of information – journal article, report, pre-print etc
- b. the main topic of the paper e.g. vaccination, attitudes etc.
- c. whether or not there was a comparison group and what it was

Thanks for the suggestions. We haven't added whether there was a comparison group as typically surveys and qualitative studies don't have comparison groups. We've added the other two columns to the table, Type of publication and Main topic of the Paper columns.

17) Table 2: Could you clarify if the surveys were cross-sectional?

Yes, this has been added to the table in new column.

18) Page 10 (Use of social media platforms as a source of information about Covid-19)–the information in this paragraph would be very nicely summarised as a graph.

This is an excellent idea. We have pulled the quantitative datasets from the evidence synthesis and summarised these percentages in a graph, linking to the appropriate text in the results section.

19) Page 10 (Use of social media platforms as a source of information about Covid-19) – did you find any information about whether the sources of information were official vs unofficial? In the UK, Public Health England and the NHS undertook social media marketing on Twitter and even on Youtube by collaborating influencers.

This is reported in this section; e.g. “A survey of 774 refugee households in Southeast Turkey⁴⁸ found the majority (75%) obtained COVID-19 information from Facebook, YouTube, Twitter ... only 10% reported getting information from NGO/UN sources.” We have updated the text to provide the full names – “Non-governmental organisations (NGO)/United Nations (UN)”. This section synthesises results about reports of social media usage (typically from surveys). There were studies about creating digital content to support health promoting behaviours through accurate information (e.g. Despres, 2020), and good practice for this is in the section ‘Good practice in promoting information and countering misinformation’.

20) Page 11 line 29: typo on member

21) Page 11 line 43: typo in first sentence

Thanks for spotting these!

22) Table 3: Again -this may need permission for reproducing. I'd suggest summarising in one paragraph rather than having a table here.

We are really keen to include this paper as it pulled some interesting qualitative data from papers included in this evidence synthesis. We have created the table from scratch, so no copyright issues, and have clearly cited the sources of information and first authors.

23) Page 15 line 51-55: I read this a few times but struggled with understanding the meaning. Consider re-phrasing/clarifying.

Thank-you. We have updated to include the words “encouraging rule-breaking behaviour through socially normalising such behaviour by enabling” – thus this section now reads “Another US study among Latino people reported that social media acted as a potential deterrent for following some public health measures to prevent infection by encouraging rule-breaking behaviour through socially normalising such behaviour by enabling people to observe the negative, guideline-breaking behaviours of others in social media posts”

24) Figure 1: the full-text articles excluded add up to 118 instead of 108. Is this because some articles were excluded for more than one reason?

Thanks for spotting, this was a typo (34, not 44 studies were excluded for the wrong outcome). This has been updated in the text.

25) Figure 2 is not necessary as it repeats the information presented in table 2. I can see you are

highlighting the lack of research in low-income countries and more generally worldwide but this is sufficiently done in the text.

We feel that Figure 2 is appealing to readers and is a nice visual representation of the geographical distribution of studies so we are keen to keep this figure in, but have moved it to supplementary materials. As noted in our discussion, we are very concerned that this type of research is only predominantly being carried out in high-income settings, especially as COVID-19 vaccine roll out is now gathering considerable momentum in LMICs.

Discussion

26) Typo on “tacking”

Thanks for spotting.

27) Please add a strengths and limitation section

There are bullet points for the strengths and limitations after the abstract, in accordance with the journal style. We have a paragraph about strengths and limitations in the discussion and the journal style does not state that this should have a subtitle.

28) Table 4 is not necessary. The key messages should be in the opening paragraph of your discussion. You can add a section to your discussion on the implications for policy and practice. Alternatively, the recommendations could work nicely as a figure/infographic.

We feel the bullet points in table 4 work really well to clearly communicate an overall summary of our key messages, we always do this for our papers and often journals request it.. We feel we have woven implications for policy and practice through the discussion, and they come out most clearly and starkly in Table 4, but happy with any editorial decision here.

Reviewer: 2

Dr. Ana Requena-Mendez, Instituto de Salud Global Barcelona, Karolinska Institute

Comments to the Author:

This article reviews the evidence about the use of social media platforms by migrant and ethnic minority communities for information about COVID-19. Also, it has tried to address the misinformation via these platforms and implications for health behaviours including vaccine intent and uptake. I think it is an interesting topic although data and studies are very scarce in this field. I also think the methodology is appropriate for a systematic review.

Thank you for reviewing and for your positive comments.

I have several general comments:

1. Authors provide information on the use of social media platforms as a source of information about COVID-19. Are there specifications about the language for which these COVID-19 information is searched (country of birth language, host-country language, English....? This would be very useful for future policy planning for example.

This was mostly not reported, and many of the social media platforms are popular in a range of languages, and the participants in some studies would speak a range of languages. For example, the top 10 languages on Twitter are English, Japanese, Portuguese, Indonesian, Spanish, Dutch, Korean, French, German and Malay. (data from { <https://www.parc.com/blog/languages-and-social-network-behaviors-top-10-languages-on-twitter/> })

We have expanded the pre-existing text: “One study exploring the views of US Latinos reported that they consulted national and local news reports for information about COVID-19 and many reported that they got their news from Spanish-language news due to difficulty in

understanding news in English; some received their news from social media sources, including Facebook, but expressed caution around messages from social media as there was no way to ensure the accuracy of the reports.⁴²”

With the text: “Language barriers were also reported in the Syrian refugee population in South-East Turkey, who typically prefer information in Arabic,⁴⁸ the Latino population in the U.S,^{37 39 41} for people from a range of ethnic groups in Bradford²⁹ and for international migrants.³²”

2. I would try to specify more in the section “use of social media platforms”, which of these media are official sources of information (e.g., twitter or facebook from public health agencies, NGOs...) and which ones are coming from “unofficial sources”. If studies do not differ this aspect, I would introduce it as a major limitation of such studies.

This is an excellent point. The studies didn’t specifically state this, but the sense was from reading across the whole paper that the sources were most definitely informal. In addition, many govts, including the UK, hadn’t used social media well in the past for public health communications, this is a field that has rapidly evolved during the pandemic (especially in terms of communicating to marginalised groups outside of health systems/marginalised from official public health comms), so we can be quite confident that these individuals discussed in the datasets were using informal channels, diaspora media, sources of information in their own language, that were not the official information from their host country. We have, however, introduced this as a limitation of the studies. However, one of the main points we want to make in the paper is that there is an opportunity for official providers of information to disseminate messages more widely through social media and to promote their presence on social media, if they have one already. We report more about this in the ‘Drivers of social media reliance’ section; for example:

“Several studies highlight concerns that some migrant and ethnic minority groups were unable to find official information in their host country in their native language about various aspects of COVID-19, hence their reliance on social media^{29 32 34 45 48}. For example, a UK study of precarious migrants (asylum seekers, undocumented migrants) reported that those feeling most abandoned or scared due to a lack of understandable, clear official information in the early stages of the pandemic were more likely to rely on word-of-mouth or social media (WhatsApp groups, Facebook) for information, including around the vaccination programme³⁴. One study of international migrants in China (94.5% of whom preferred social media for news about COVID-19) had lower rates of correct knowledge about COVID-19 compared to rates reported for Chinese residents⁴⁵. The authors speculate that this might be due to a lack of available public health information in a range of languages.”

We have also added the following:

“We also acknowledge that as many of the surveys didn’t formally report whether the social media feeds their responders were following were from ‘official’ sources, such as government or non-governmental organisations or from ‘unofficial’ sources, such as friends, relatives, or accounts simply with many followers, though the included datasets and further qualitative work our groups is currently doing in the UK suggest they will be predominantly unofficial sources, with government public health teams in several countries very slow to make effective use social media as a platform of communication at the start of the pandemic.

3. Are there any differences considering the type of migrants (refugees/asylum seeker, undocumented migrants...)? Authors recognize the limitation concerning the availability and quality of the datasets and also that that migrants and ethnic minorities are a highly diverse group with a range of health and socioeconomic situations making it hard to generalize. However, authors should compare different migrant groups, in particularly regarding the access to official information in their host country but also on the “social media impact on preventative health measures and vaccine intent”. Add it otherwise as a limitation.

We have added “A further limitation is that there were insufficient studies to reliably compare use of social media across type of migrants (refugees/asylum seeker, undocumented migrants), and future research should explore this.”

4. Authors highlight the stark lack of data from low and middle-income countries. What are possible explanations for that? Publication bias? Are there other databases /search strategy (including non-scientific publications) that could add the perspective or context of low-middle income countries?.

We did an extremely comprehensive search across multiple databases and spent a lot of time exploring the grey literature, so we are highly confident that we have not missed anything. We have added “It may be that the lower availability of research funding in low-income countries may explain the lack of studies from these countries, as well as the fact that this is a very new and evolving ” to the text.

Minor comments:

Page 13, Line 43. “Studies reported that some migrant ...” which studies? Please specify and also specify the targeted population

We’ve added this.

Page 21, line 20. Why are not these recent studies not included in the review?
I suggest updating the search strategy and to include reference 47-50 and 52.

These papers were found after the search had concluded and the search would not be comprehensive if we extended the dates to just include this study. We have published multiple systematic reviews and any relevant papers that come up after the formal search concludes, quality appraisal done/data extraction duplicated etc, we put in the Discussion so they are included.

Page 23, line 29. “There is a stark lack of data on social media use from low and middle-income countries...”. I agree, but also from high income countries. Actually, the majority of studies were focused on USA and UK which also limits the generalization of the results also for migrant in high income countries. I would highlight it as a limitation of the study.

Good point. We’ve added “Studies from high income countries are also limited, with the majority of studies focused on the USA and UK.”

VERSION 2 – REVIEW

REVIEWER	Requena-Mendez, Ana Instituto de Salud Global Barcelona
REVIEW RETURNED	31-Aug-2022
GENERAL COMMENTS	I thank the reviewers the answers provided to all my comments. I think the manuscript has significantly improved and it deserves being accepted for publication